# Highly Glycolytic Immortalized Human Dermal Microvascular Endothelial Cells are Able to Grow in Glucose-Starved Conditions

**DOI:** 10.3390/biom9080332

**Published:** 2019-08-01

**Authors:** Mª Carmen Ocaña, Beatriz Martínez-Poveda, Ana R. Quesada, Miguel Ángel Medina

**Affiliations:** 1Departamento de Biología Molecular y Bioquímica, Facultad de Ciencias, Andalucía Tech, Universidad de Málaga, E-29071 Málaga, Spain; 2IBIMA (Biomedical Research Institute of Málaga), E-29071 Málaga, Spain; 3CIBER de Enfermedades Raras (CIBERER), E-29071 Málaga, Spain

**Keywords:** endothelial cells, metabolism, glycolysis, lactate, MCT1

## Abstract

Endothelial cells form the inner lining of blood vessels, in a process known as angiogenesis. Excessive angiogenesis is a hallmark of several diseases, including cancer. The number of studies in endothelial cell metabolism has increased in recent years, and new metabolic targets for pharmacological treatment of pathological angiogenesis are being proposed. In this work, we wanted to address experimental evidence of substrate (namely glucose, glutamine and palmitate) dependence in immortalized dermal microvascular endothelial cells in comparison to primary endothelial cells. In addition, due to the lack of information about lactate metabolism in this specific type of endothelial cells, we also checked their capability of utilizing extracellular lactate. For fulfilling these aims, proliferation, migration, Seahorse, substrate uptake/utilization, and mRNA/protein expression experiments were performed. Our results show a high glycolytic capacity of immortalized dermal microvascular endothelial cells, but an early independence of glucose for cell growth, whereas a total dependence of glutamine to proliferate was found. Additionally, in contrast with reported data in other endothelial cell lines, these cells lack monocarboxylate transporter 1 for extracellular lactate incorporation. Therefore, our results point to the change of certain metabolic features depending on the endothelial cell line.

## 1. Introduction

Angiogenesis is the process of formation of new blood vessels from the pre-existing vascular bed [1]. In an adult organism, the angiogenic process is activated only under certain physiological conditions, and it is highly regulated by a complex network of pro- and anti-angiogenic signals. However, excessive pathological angiogenesis has been described in some diseases such as diabetes mellitus, rheumatoid arthritis, several retinopathies, and cancer [2]. Targeting angiogenesis emerged as a potential therapy for these diseases alone or in combination with other pharmacological targets, either by the use of multitargeted drugs or through the combination of several drugs targeting different pathways involved in the angiogenic process [3,4]. The study of endothelial cell (EC) metabolism has grown in importance since the discovery that a glycolytic enzyme, phosphofructokinase-2/fructose-2,6-bisphosphatase 3 (PFKFB3), is essential for vessel sprouting [5]. Glutamine was seen to be essential for the angiogenic switch as well [6]. In addition, inhibition of fatty acid metabolism, particularly carnitine palmitoyltransferase 1a (CPT1a), was also shown to disrupt angiogenesis [7]. Therefore, the metabolism of ECs has been considered a potential target for the therapy of angiogenesis-related diseases [8,9].

Despite the fact that ECs have been described as highly glycolytic cells, not many studies have tested how the presence of different metabolic substrates affects long-time dependence for proliferation and short-time uptake and utilization of others fuels in these cells. The microvasculature constitutes the vast majority of the human vascular compartment, and most pathological events take place at this level [10]. For that reason, the first objective of this work is to explore the metabolic dependence for cell growth and survival in human dermal microvascular ECs (HMEC). For this aim, we incubated the cells in different nutritional conditions for several days, which we consider to be a long-time dependence experiment. On the other hand, we wanted to confirm whether HMEC also have a high glycolytic rate, since there is at least one work that reported results that differed from the established glycolytic capacity of ECs [11]. Additionally, we studied the short-time preference for different metabolic substrates, alone or in combination, in HMEC. Ideally we would have performed these experiments after a few seconds or minutes of substrate withdrawal. However, due to experimental limitations, we increased the time to 30 min, which we consider a short time in comparison to other experiments performed after several hours. These short-time studies would allow us to look into the immediate preference for each of the metabolic fuels present at physiological concentrations in the modified media [12,13]. However, it is important to take into account that other metabolites not added to the media could be present in an individual. The purpose of not adding them to our experimental approach is to see the direct effect of the main metabolic substrates on others.

In the tumor microenvironment, cancer cells secrete large amounts of lactate [14]. Extracellular lactate exerts different effects, such as promoting immunosuppression, invasion, and angiogenesis [15]. Regarding induction of angiogenesis, lactate activates this process through different mechanisms: (1) inducing interleukin-8 (IL-8) expression in a nuclear factor kappa B (NF-κB)-dependent manner, (2) increasing vascular endothelial growth factor (VEGF), its receptor VEGFR2 and fibroblast growth factor (FGF-2) expression through hypoxia-inducible factor 1α (HIF-1α) stabilization, and (3) promoting Akt phosphorylation, all of these effects given after its incorporation via monocarboxylate transporter 1 (MCT1 transporter) [16,17,18]. However, as far as we are aware, evidence of MCT1 expression in microvascular ECs existed only in brain and eye tissues [19,20]. For that reason, our second objective is to look into lactate metabolism in dermal microvascular ECs. The results herein demonstrate a highly glycolytic capacity of HMEC, whereas the capacity of these cells to incorporate extracellular lactate is questioned due to a lack of MCT1 transporter expression.

## 2. Materials and Methods 

### 2.1. Materials

MCDB-131 cell culture medium was obtained from Gibco (Paisley, Scotland, UK). Other cell media, penicillin, streptomycin and amphotericin B, and trypsin were purchased from BioWhittaker (Verviers, Belgium). Fetal bovine serum (FBS) was purchased from Biowest (Kansas, USA). 2-NBDG (2-(N-(7-nitrobenz-2-oxa-1,3-diazol-4-yl)amino)-2-deoxyglucose) and BODIPY FL C_16_ (4,4-difluoro-5,7-dimethyl-4-bora-3a,4a-diaza-s-indacene-3-hexadecanoic acid) were supplied by Molecular Probes (Eugene, OR, USA). L-[^14^C(U)]-Glutamine was acquired from Perkin Elmer (Waltham, Massachusetts, USA). Lactate assay kit was from Abnova (Taoyuan City, Taiwan). Anti-MCT1 antibody was from Santa Cruz Biotechnology (Heidelberg, Germany), anti-HIF-1α antibody was from BD Biosciences (Franklin Lakes, NJ, USA) and anti-LC3B and anti-α-tubulin antibodies were from Cell Signaling Technology (Danvers, MA, USA). Plastic material for cell culture was from Nunc (Roskilde, Denmark). All other reagents not listed on this section, including sodium palmitate, were from Sigma-Aldrich (St. Louis, MO, USA).

### 2.2. Cell Culture

Cell culture media were supplemented with glutamine (2 mM), penicilin (50 U/mL), streptomycin (50 U/mL), amphotericin (1.25 μg/mL), and 10% fetal bovine serum (FBS) unless specified otherwise. Immortalized human microvascular endothelial cells (HMEC) were kindly supplied by Dr. Arjan W. Griffioen (Maastricht University, Netherlands), maintained in MCDB-131 medium supplemented with hydrocortisone (1 μg/mL) and epidermal growth factor (EGF) (10 ng/mL) and generated as described [10]. Bovine aortic endothelial cells (BAEC) were isolated from bovine aortic arches as previously described and maintained in Dulbecco’s modified Eagle’s medium (DMEM) containing glucose (1 g/L) [21]. Human umbilical vein endothelial cells (HUVEC) were isolated by a modified collagenase treatment, as previously reported and maintained in endothelial cell growth basal medium (EBM-2 medium) supplemented with 10% fetal bovine serum and EGM-2 SingleQuots [22]. Transformed human HL-60 leukemia cells and human cervix adenocarcinoma HeLa cell line were purchased from the ATCC (Rockville, MD, USA) and maintained in RPMI with 20% FBS and Eagle’s minimum essential medium (EMEM), respectively. Cells were maintained at 37 °C under a humidified 5% CO_2_ atmosphere. For hypoxic conditions, 200 µM CoCl_2_ for 24 h was added to the cells and incubated in a normal atmosphere, or cells were incubated in an incubator with 1% O_2_.

### 2.3. Conjugate Palmitate-BSA Preparation

Sodium palmitate was conjugated with bovine serum albumin (BSA) essentially fatty acid free at 5:1 proportion according to a published protocol [23].

### 2.4. Cytotoxicity Assays

HMECs were seeded at a density of 2 × 10^4^ cells in 96-well plates, and after 24 h 5 mM glucose, 0.5 mM glutamine and/or 0.5 mM palmitate were added for another 6 h. After incubation, 3-(4,5-dimethylthiazol-2-yl)-2,5-diphenyltetrazolium bromide (MTT) was added to the wells and incubated for 4 h. Absorbance was read at 550 nm with an Eon Microplate Spectrophotometer from Bio-Tek Instruments (Winooski, VT, USA). Data was collected by Gen5 software (version 2.05) from the same manufacturers. Four samples for every tested concentration were included in each of three independent experiments.

### 2.5. Cell Growth Curves

HMEC, BAEC, HUVEC, and HeLa were seeded at a density of 1.5 × 10^4^ cells in 24-well plates and after cell adherence media were changed and different combinations of glucose, glutamine, and/or pyruvate were added. Cells were incubated in normoxia or in hypoxia (1% O_2_) along with 5 mM HEPES. Cells were counted every day for 5 days by using a Coulter counter from Beckman Coulter (Brea, CA, USA).

### 2.6. EdU Proliferation Assay

HMECs were seeded and exposed to glucose and/or glutamine withdrawal when they were in an exponential growth phase. After 22 h, 10 µM EdU was added to each well for additional 2 h. Cells were harvested and EdU incorporation was detected using a baseclick EdU Flow Cytometry Kit (Baseclick GmbH) in a FACS VERSE^TM^ cytometer from BD Biosciences (San Jose, CA, USA) and data were analyzed with its software BD FACSuite (version 1.0.5.3841).

### 2.7. Wound Healing Migration Assay

Cells were seeded in 6-well plates and grown until confluence. Then, a scratch was made in the confluent monolayer using a sterile 200 µL pipette tip, cells were washed with PBS, and media were changed to DMEM lacking glucose and/or glutamine. Cells were incubated and photographs were taken after 0, 4, and 7 h. Images were analyzed with ImageJ software (version 1.50). Wound closure was calculated as the percentage of the initial wounded area (time 0) that had been recovered by migrating ECs after 4 or 7 h.

### 2.8. Flow Cytometry Assay for Cell Cycle Distribution

HMECs were seeded at a density of 7.5 × 10^4^ cells in 6-well plates and after cell adherence medium was changed and different combinations of glucose and glutamine were added. After 48 h incubation, attached and unattached cells were harvested, washed with PBS, and fixed with 70% ethanol for 1 h on ice. Pelleted cells were incubated with 0.1 mg/mL RNase-A and 40 µg/mL propidium iodide at 37 °C for 30 min protected from light. Percentages of subG1, G0/G1 and S/G2/M populations were determined using a FACS VERSE^TM^ cytometer from BD Biosciences (San Jose, CA, USA), and data were analyzed with its software BD FACSuite (version 1.0.5.3841).

### 2.9. Extracellular Flux Analyzer Experiments

HMEC were cultured at 3 × 10^4^ cells at 24-well Seahorse XF^e^24 plates (Agilent) and were incubated at 37 °C under a humidified 5% CO_2_ atmosphere overnight. Cells were washed twice with XF base medium (Agilent) and incubated with XF base medium at 37 °C without CO_2_ for one hour. Three initial measurements were made using XF^e^24 Seahorse analyzer, and then 5 mM glucose, 0.5 mM glutamine, 0.5 mM palmitate, 10 mM lactate, or combinations of them were injected and five additional measurements were performed. Data were analyzed with Wave software (version 2.6.0).

### 2.10. FACS Analysis of Glucose and Palmitate Uptake

Cells cultured in 96-well plates or 24-well plates were washed twice with PBS supplemented with calcium and magnesium (DPBS), and then starved for 30 min with this DPBS. For the glucose uptake assay, cells were incubated for additional 30 min with DPBS supplemented with 5 mM glucose and 0.5 mM glutamine and/or 0.5 mM palmitate and 100 µM 2-NBDG. For the palmitate uptake assay, cells were incubated for an additional 30 min with DPBS supplemented with 0.5 mM palmitate and 5 mM glucose and/or 0.5 mM glutamine and 2 µM BODIPY FL C_16_. Relative glucose or palmitate uptake was determined using a FACS VERSE^TM^ cytometer from BD Biosciences (San Jose, CA, USA) as previously described [24,25]. Data were analyzed with BD FACSuite software (version 1.0.5.3841).

### 2.11. Glutamine Consumption and Oxidation

Cells cultured in 96-well or 24-well plates were washed twice with PBS supplemented with calcium and magnesium (DPBS), and then starved for 30 min with this DPBS. Then, cells were incubated for additional 30 min with DPBS supplemented with 25 mM HEPES, 0.5 mM glutamine and 5 mM glucose and/or 0.5 mM palmitate and 0.5 µCi/mL L-[^14^C(U)]-glutamine. For glutamine consumption, media were collected and mixed with scintillation liquid. For glutamine oxidation, media and cells were collected in round-bottom glass tubes with screw-caps. Each glass tube contained a Whatman^TM^ paperfold imbibed with benzethonium hydroxide (hyamine). 400 µL 10% (*v*/*v*) HClO_4_ were added to the closed glass tubes through the cap. Tubes were incubated for an additional 30 min at 37 °C with agitation. Whatman^TM^ paperfolds with ^14^CO_2_ captured in them were mixed with scintillation liquid. A Beckman Coulter LS6500 liquid scintillation counter (Fullerton, CA, USA) was used for the measurements. All assays were performed in the Radioactive Installation of the University of Málaga, authorized with reference IR/MA-13/80 (IRA-0940) for use of non-encapsulated radionuclides.

### 2.12. Lactate production

For lactate production measurement, a modified version of the protocol described previously was used [26]. Cells cultured in 6-well plates were washed twice with PBS supplemented with calcium and magnesium (DPBS), and then starved for 30 min with this DPBS. Then, cells were incubated for an additional 30 min with DPBS supplemented with 5 mM glucose, 0.5 mM glutamine, and/or 0.5 mM palmitate. Media were collected, deproteinized with 10% (*v*/*v*) HClO_4_ (5:1), and neutralized with 20% (*w*/*v*) potassium hydroxide (KOH). Lactate was measured using a lactate assay kit following manufacturer instructions. A FL600FA fluorescence microplate reader from Bio-Tek Instruments (Winooski, VT, USA) was used. Data were collected using KC4 software (version 2.5) from Bio-Tek Instruments with Ex/Em = 535/590 nm.

### 2.13. RNA Isolation and Purification and cDNA Synthesis

Cells were seeded in 6-well plates under different conditions. Then they were harvested in Tri reagent and total RNA was isolated with the Direct-zol™ RNA MiniPrep Kit (Zymo Research) according to the purchaser’s instructions. RNA amount and quality (260/280 ratio) were measured using a NanoDrop ND-1000 from Thermo Scientific.

1 µg of RNA was converted to complementary DNA (cDNA). The High-Capacity cDNA Reverse Transcription Kit (Applied Biosystems) was used for cDNA synthesis according to the purchaser’s instructions.

### 2.14. qPCR

For quantitative RT-PCR (qPCR), total RNA isolation and complementary DNA synthesis were performed as described above and PCR reactions were done using KAPA SYBR Fast Master Mix (2×) Universal (KAPA Biosystems) in an Eco Real-Time PCR System. The following thermal cycling profile was used: 95 °C, 3 min; 40 cycles of 95 °C, 10 s; annealing temperature (Tm), 30s. qPCR was performed in duplicate for each sample of three different experiments in keeping with the manufacturer instructions. All qPCR data were normalized to β-actin expression. Primers sequence, Tm, and amplicon size for each gene are shown in Table 1.

### 2.15. Western Blot

Cells were lysed with radioimmunoprecipitation assay buffer (RIPA buffer) or 2× denaturing loading buffer. Samples were heated and separated on 10% or 15% polyacrylamide gels. Proteins were transferred to nitrocellulose membranes and blocked with 10% (*w*/*v*) semiskimmed dried milk. Blocked membranes were incubated overnight with primary antibodies (LC3B 1:1000, MCT1 1:100, HIF-1α 1:500, α-tubulin 1:10000), washed, and later incubated with the peroxidase-linked secondary anti-rabbit or anti-mouse antibody for 1 h at room temperature. Membranes were washed and finally incubated with the Supersignal^®^ West Pico chemiluminescent substrate system (Thermo Scientific). Image captions were taken with the ChemiDoc^TM^ XRS+ System (Bio-Rad) and densitometry analyses were made with Image Lab^TM^ software (version 6.0).

### 2.16. Statistical Analysis

Results are expressed as mean ± SD. Statistical significance was determined using the two-sided Student *t*-test or one-way ANOVA with post hoc test. Values of *p* < 0.05 were considered to be statistically significant.

## 3. Results

### 3.1. Glutamine, but not Glucose, is Essential for HMEC Growth

For this work, we first wanted to test the growth of HMEC under different nutritional conditions. However, this experiment could not be performed with palmitate since this long chain fatty acid is toxic to HMEC at 0.5 mM as soon as after 6 h incubation (Figure 1a). In order to see the dependence of HMEC on glucose and glutamine, cells were seeded at a low concentration and exposed to combinations of glucose and/or glutamine for five days. HMEC were able to grow in the absence of glucose for the first three days at the same rate as cells grown with both glucose and glutamine (Figure 1b). However, HMEC did not grow under glutamine starvation even in the presence of glucose (*p* < 0.05) (Figure 1b). This did not happen in a macrovascular endothelial cell line such as BAEC or in a tumor cell line such as cervix adenocarcinoma (HeLa) (Appendix A). Growth curves of the human macrovascular endothelial cell line HUVEC could not be determined due to their strict culture conditions, leading cells to death after day 1 (Appendix A).

Importantly, these experiments were performed in a different medium (DMEM) than the growth medium that HMEC are cultured with (MCDB-131). Growth rate was lower in DMEM as compared to growth medium (Figure 1c). One major difference between these media is the presence or not of sodium pyruvate. Thus, an additional experiment was performed in the presence and absence of pyruvate along with glucose and/or glutamine. Pyruvate increased growth rate in all conditions, although it was only statistically significant in the condition without glutamine (*p* < 0.05) (Figure 1c). Another important difference is glutamine concentration. MCDB-131 was supplemented with 2 mM glutamine, whereas our DMEM was supplemented with the physiological concentration of 0.5 mM glutamine. However, increasing glutamine up to 2 mM in DMEM did not improve growth rate in HMEC (Appendix A).

On the other hand, endothelial cells often confront hypoxia. For that reason, HMEC were also grown in the presence or absence of glucose and glutamine under hypoxia. Glucose starvation still allowed cells to grow in the presence of glutamine as compared to the ones grown in the presence of glucose and glutamine, but to a lesser extent than in normoxia (*p* < 0.05) (Appendix A).

Additionally, cell proliferation was also determined by means of an EdU proliferation assay. In the absence of glutamine, proliferating cells were almost inexistent (3% of the total population in the best scenario) (*p* < 0.01), whereas glucose starvation did not affect proliferation after 24 h of incubation (Figure 2).

### 3.2. Glucose Starvation Does Not Affect Cell Migration, Cell Cycle Distribution and Does Not Induce Autophagy

Besides proliferation, cell migration is another stage involved in angiogenesis. Due to the early independence of HMEC of glucose to grow, we wanted to check if migratory capacity of these cells was affected after glucose withdrawal. Our results show that glucose starvation does not affect cell migration of HMEC (Figure 3). The migratory capacity of HMEC was also tested under glutamine withdrawal. Cells barely migrated in these conditions (Figure 3). However, after 7 h incubation in the absence of glutamine, cells started to detach, and hence the role of glutamine in cell migration under these conditions should be carefully evaluated.

In order to see cell cycle distribution under glucose starvation, we performed a cell cycle analysis by flow cytometry. Glucose withdrawal for 48 h neither altered cell cycle nor induced subG1 accumulation, an indicator of apoptosis, if glutamine was present (Figure 4a).

Next, we wanted to analyze if glucose starvation induces autophagy in HMEC. Cells were grown in the presence or absence of 5 mM glucose for 48 h, and LC3B-II/LC3B-I protein expression was determined by Western blot. Glucose starvation did not increase the LC3B-II/LC3B-I ratio in these cells after 48 h. Cells incubated with 5 mM glucose and 50 µM chloroquine for 16 h were used as positive control (Figure 4b).

### 3.3. Oxygen Consumption and Extracellular Acidification Rates in HMEC

Oxygen consumption rate (OCR) and extracellular acidification rate (ECAR), indicators of oxidative phosphorylation (OXPHOS) and glycolysis, respectively, were measured in HMEC using an XF^e^24 Seahorse Flux Analyzer. Three basal measurements were made in the absence of glucose, glutamine, or palmitate, and then these metabolic substrates were injected into the wells. A slight OCR increase was detected in all cases (*p* < 0.05) (Figure 5a). On the other hand, ECAR increased to more than 300% of the basal measurement in the presence of glucose (*p* < 0.001) (Figure 5b), indicating a high glycolytic capacity of these cells.

### 3.4. Metabolic Fuel Uptake and Metabolism in HMEC

Uptake of three different metabolic fuels (glucose, glutamine and palmitate) after 30 min incubation was measured using fluorescent analogues or radiotracers. Regarding glucose, neither glutamine nor palmitate affected glucose uptake in the conditions tested (Figure 6a). However, palmitate increased lactate production in the presence of glucose (*p* < 0.01) (Figure 6b). Lactate was not detected in the media in the absence of glucose (Figure 6b). On the other hand, glucose did not diminish glutamine uptake statistically (Figure 6c), but it reduced glutamine oxidation as represented by measurement of the released ^14^CO_2_ (*p* < 0.01), and this effect was lower when palmitate was also added to the media (*p* < 0.05) (Figure 6d). With respect to palmitate uptake, neither glucose nor glutamine affected it (Figure 6e).

We also studied glucose and glutamine uptake, as well as lactate production and glutamine oxidation, in the presence of glucose and/or glutamine in HUVEC. Similar results to those of HMEC were obtained (Figure 7).

### 3.5. Lactate Metabolism in HMEC

The capacity of microvascular endothelial cells for oxidizing lactate was tested by means of XF^e^24 Seahorse Flux Analyzer. The results indicated that lactate at a concentration of 10 mM is not oxidized by HMEC since OCR did not increase in the presence of this metabolite (Figure 8a). Moreover, presence of lactate blocked the increase in ECAR in the presence of glucose (*p* < 0.01) (Figure 8b). We then wanted to know if HMEC lacked some of the enzymes and/or transporters involved in lactate metabolism. For that purpose, mRNA expression of LDH-A, LDH-B, MCT1 and MCT4 were measured. No mRNA expression was detected for either LDH-B or MCT1 (Figure 8c). Afterwards, since hypoxic conditions are usually given in the angiogenic microenvironment, we performed these experiments under hypoxia. To this effect, we added 200 µM CoCl_2_ to the media in cells cultured in normoxia so anoxia was induced, or cells were cultured under 1% O_2_ atmosphere. The major effect on HIF-1α protein expression was under the hypoxia condition (Appendix A). MCT4 mRNA expression was greater in CoCl_2_ and hypoxia as compared to normoxia condition (Appendix A). MCT1 mRNA expression could not be detected in anoxia or hypoxia (Appendix A). Furthermore, MCT1 protein was not detected in any condition either (Figure 8d). Additionally, 10 mM lactate failed to induce MCT1 protein expression (Figure 8e). HL-60 was used as positive control for MCT1 Western blot (Figure 8d,e) [27].

## 4. Discussion

Study of EC metabolism and its relationship with angiogenesis has been an emerging topic in recent years, and several reviews have been published regarding this topic [9,28,29,30]. Most of the studies have focused the attention on gene silencing or long-time incubations with inhibitors of different key steps of the main metabolic pathways in order to see the effect on the angiogenic process [5,6,7,31,32]. These publications highlighted the importance of glucose, glutamine, and fatty acid metabolism on angiogenesis. However, non-physiological concentrations were used in most of these works, which would affect EC metabolism [33,34]. For that reason, physiological concentrations (5 mM glucose, 0.5 mM glutamine and 0.5 mM palmitate) were used in the experiments herein performed [13].

For this work, we first wanted to see HMEC growth dependence on different metabolic fuels. However, palmitate is known to induce apoptosis in several cell lines in culture, including both macrovascular and microvascular ECs [35,36,37,38,39]. Accordingly, palmitate was also found to decrease cell viability in HMEC. For this reason, we could not perform cell growth experiments with palmitate since it would need to be present in the media for periods of time in which toxicity would be too high. Additionally, although fatty acids have been shown to be essential for vessel sprouting, it has been observed that palmitate inhibits cell invasion in ECs, a limiting step in angiogenesis [7,40]. Therefore, we continued only with glucose and glutamine for long-period experiments.

In this work, we found that HMEC are able to sustain cell growth in the absence of glucose for at least the first three days of culture. After that period of time, glucose becomes necessary for maintaining cell growth. However, glutamine was found to be completely essential for cell proliferation. Indeed, glucose deprivation does not alter either cell cycle distribution or proliferation, whereas the absence of glutamine drastically diminishes cell proliferation. Moreover, glucose starvation does not induce apoptosis in these cells since no accumulation of cells is found in subG1 under these conditions, whereas others found that glucose withdrawal for 24 h increased caspase 3 levels in retinal capillary ECs [41]. Additionally, it should be taken into account that HMEC growth in DMEM was lower compared to cell growth in MCDB-131. Addition of pyruvate to the media or increasing glutamine concentration did not improve cell growth. In contrast, increasing glutamine concentration from 0.5 mM to 2 mM improved proliferation of HUVECs [42]. Remarkably, MCDB-131 media contains additional metabolites not present in DMEM which could improve cell growth of HMECs, namely alanine, asparagine, aspartate, proline, biotin, vitamin B12, adenine, lipoic acid, putrescine, and thymidine. On the other hand, since ECs often confront hypoxia conditions, we also tested glucose and glutamine dependence for cell growth under hypoxia. Our data show that glutamine by itself is able to sustain cell growth under hypoxia in the absence of glucose, although to a lesser extent than in normoxia. It has been shown that under hypoxic conditions glutamine can still be oxidized by tumor cells in order to sustain cell metabolism [43,44]. All in all, HMECs seem to be completely dependent on glutamine, but not glucose, to grow. Remarkably, our HMECs are an immortalized cell line with a faster growth rate and less strict nutritional requirements than primary ECs (i.e., they are able to grow in lower serum concentrations in culture compared to primary microvascular ECs) [10]. The use of immortalized ECs offers several advantages compared to primary microvascular ECs. However, it should be taken into account that results from proliferation or apoptosis assays should be carefully interpreted when this kind of cells are used [45]. If this lower dependence on glucose to grow compared to other ECs is due to HMEC being a microvascular cell line or whether their immortalization is involved remains to be further researched.

On the other hand, it is known that under nutrient depletion conditions autophagy may be induced in order to protect cells from starvation and assure cell survival [46]. For instance, glucose starvation in mouse microvascular ECs, as well as inhibition of glucose metabolism with 2-deoxyglucose in HUVEC, has been shown to induce autophagy [47]. In contrast, our data show that glucose deprivation in HMEC does not affect the LC3B-II/LC3B-I ratio, an indicator of autophagy. Thus, these cells seem to be able to grow and survive in the absence of glucose, most likely by means of a mechanism different from autophagy. Other authors found that ECs in the absence of glucose are able to alter their metabolism to survive, by different mechanisms involving or not glycogenolysis [48,49,50]. Further research should be performed in order to determine the specific molecular mechanisms triggered by glucose starvation in HMEC.

It is known that both glucose and glutamine are essential for tube formation [5,6,42]. However, different results have been obtained for migration experiments performed under glucose or glutamine withdrawal [6,42]. The results obtained using HMEC showed that glucose is not essential for cell migration in this cell line. This is in contrast with previous results obtained with HUVEC [42]. However, this may be in line with the different regulation of long-time glucose metabolism found in HMEC compared to other cell lines. On the other hand, cell migration was drastically reduced in the absence of glutamine. Noticeably, after 7 h incubation under glutamine starvation cells began to detach. Therefore, it may be possible that glutamine withdrawal does not affect specifically cell migration in HMEC but cell function in general.

During the last century, uptake and utilization of different metabolic substrates were tested in the presence of these different metabolic fuels alone or in combination, using tumor cell lines or healthy cell lines [51,52,53,54,55]. Nevertheless, few studies were carried out under these conditions in ECs [11,56,57,58]. Moreover, incubation times as long as six days have sometimes been used [59]. In this work, we wanted to test the direct influence of different metabolic substrates on the uptake and utilization of the others after a short period of time. For extracellular flux analyses, basal measurements were performed in the absence of the main metabolic substrates, but it should be taken into account that the basal media contained essential amino acids and other molecules. For the rest of the experiments, a first fast period of 30 min was induced in DPBS so that no metabolic fuel was present and cells would respond avidly to addition of the substrates for another 30 min. Ideally, these experiments should have been performed after a few seconds or minutes in order to observe the immediate preference of the cells, but the methodological procedures used were not sensitive enough for detecting small changes after such very short incubation times. Under the conditions finally used, we found that HMEC slightly oxidize glucose, glutamine, and palmitate. On the other hand, due to the high ECAR increase after glucose addition, together with the small increase in OCR after glucose or glutamine addition, it is likely that HMEC preferably converts glucose to lactate instead of oxidizing glucose or glutamine. Moreover, the presence of glucose decreases glutamine oxidation in HMEC and also in HUVEC, which reinforces the fact that these cells are highly dependent on glycolysis. This corresponds with most of the data available in the bibliography [5,56,60,61]. No extracellular lactate was detected in the absence of glucose, although ECAR was increased in the presence of glutamine alone. This can be explained by deprotonation of HCO_3_^-^ resulting from oxidation [62]. On the other hand, no major differences were found on glucose and palmitate uptake in the presence of the other metabolic substrates. The similar results found in HUVEC for this short-time substrate preference indicates a similar preference for metabolic substrate between these two models of microvascular and macrovascular ECs. As a conclusion, our model of microvascular ECs is highly glycolytic and glucose represses glutamine metabolism. This fact corroborates how cells can rely on different metabolites depending on their necessities. HMEC probably obtain more energy from glucose via glycolysis, like many tumor cells do, whereas they need glutamine most likely in order to synthesize acetyl-CoA for lipid synthesis to sustain cell growth.

In the tumor microenvironment, tumor cells secrete great amounts of lactate to the media. It is known that this lactate could be taken up by stromal cells. For example, cancer-associated fibroblasts (CAFs) use this lactate to obtain energy via oxidation [63]. Regarding ECs, they have been shown to incorporate lactate from tumor cells via MCT1 transporters [16]. However, unlike CAFs, ECs do not oxidize this lactate at a major extent when glucose is present [56]. In our experimental model, we observed that HMEC could not oxidize lactate even in the absence of glucose. Lactate taken up by ECs is known to induce angiogenesis without being oxidized, acting on the NF-κB/IL-8 pathway, avoiding HIF-1α degradation through prolylhydroxylase (PHD) inhibition and activating the PI3K/Akt pathway, thus triggering the angiogenic process [16,17,18]. Targeting MCT1 transporter or lactate dehydrogenase B (LDH-B) enzyme reverses the effect to the one observed in the control without lactate [16,17]. Since all of these experiments were usually carried out using macrovascular ECs and no MCT1 expression has been reported so far in dermal microvascular ECs, we wanted to verify the presence of MCT1, as well as that of LDH-B, in HMEC. No mRNA expression was detected either for MCT1 or for LDH-B. Reproducing the characteristic hypoxic condition present in the tumor microenvironment did not induce the mRNA or protein expression of MCT1 either, whereas MCT4 expression was increased. However, it is known that MCT1 does not have hypoxia response elements (HRE) at its promoter [64]. Moreover, the presence of lactate did not induce MCT1 protein expression either. MCT1 is the main and most characterized transporter described for lactate uptake. Other less characterized lactate transporters, such as MCT2, have been reported in certain tissues [65]. The presence of these transporters in ECs has not been described and further investigation should be performed in future works. The lack of expression of MCT1 magnifies the fact that not all ECs, and more specifically not all microvascular ECs, behave in the same way, and that a model of ECs is not representative of all kinds of ECs [66,67]. Tissue origins, as well as the kind of blood vessel the ECs come from, may influence the metabolism of isolated cells. It has been demonstrated that targeting MCT1 can inhibit tumor angiogenesis in vivo [17]. However, the results presented herein suggest the limitations of this target. Thus, it seems likely that targeting glycolytic enzymes may offer a better strategy for the treatment of angiogenesis-dependent diseases, which could certainly be improved with additional MCT1 targeting in some tissues such as those in the brain and retina [20].

## 5. Conclusions

Microvascular endothelial cells have been less studied compared to macrovascular endothelial cells. Their laborious isolation, as well as usual dependency, on human serum makes them non-friendly cells for reproducible results. In this work, the use of an immortalized microvascular endothelial cell line allowed us to consistently demonstrate their short and long-time dependency on glucose and glutamine metabolism, respectively. Furthermore, whereas MCT1 is highly expressed in HUVEC and endothelial cells from the brain and retina, this lactate-H^+^ transporter is not present in our model of dermal microvascular endothelial cells. Although the short-time preference was similar between HMEC and HUVEC, results from MCT1 expression suggest that different endothelial cells can behave in a different way most likely depending on the kind of capillary they were isolated from, as well as the tissue origin. Strategies for the treatment of angiogenesis-dependent disorders should always be designed according to these differences.

## Figures and Tables

**Figure 1 biomolecules-09-00332-f001:**
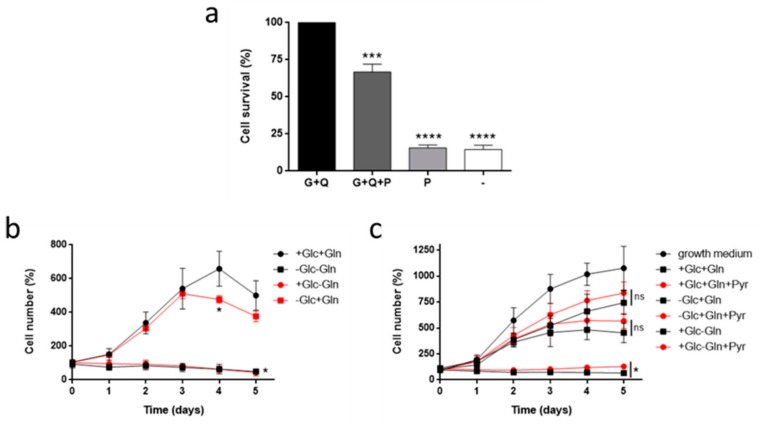
HMEC growth under different nutritional conditions. (**a**) Cell survival after 6 h incubation in DMEM supplemented with 5 mM glucose (G), 0.5 mM glutamine (Q) and/or 0.5 mM palmitate (P). (**b**) HMEC growth was monitored in the presence or absence of 5 mM glucose and/or 0.5 mM glutamine and (**c**) 5 mM glucose and/or 0.5 mM glutamine with and without 1 mM sodium pyruvate. Data are expressed as means ± SD of three independent experiments. **p* < 0.05, ****p* < 0.001, *****p* < 0.0001 versus glucose and glutamine condition (**a**,**b**) or condition without pyruvate (**c**).

**Figure 2 biomolecules-09-00332-f002:**
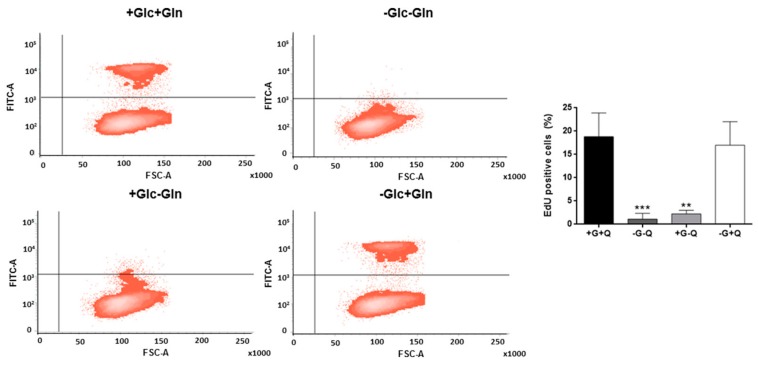
HMEC proliferation under different nutritional conditions. 10 μM EdU was added for 2 h to cells already incubated for 22 h in the presence or absence of 5 mM glucose (G) and/or 0.5 mM glutamine (Q). EdU incorporation was detected using a FACS VERSE^TM^ cytometer. Data are expressed as means ± SD of three independent experiments. ***p* < 0.01, ****p* < 0.001 versus condition with glucose and glutamine.

**Figure 3 biomolecules-09-00332-f003:**
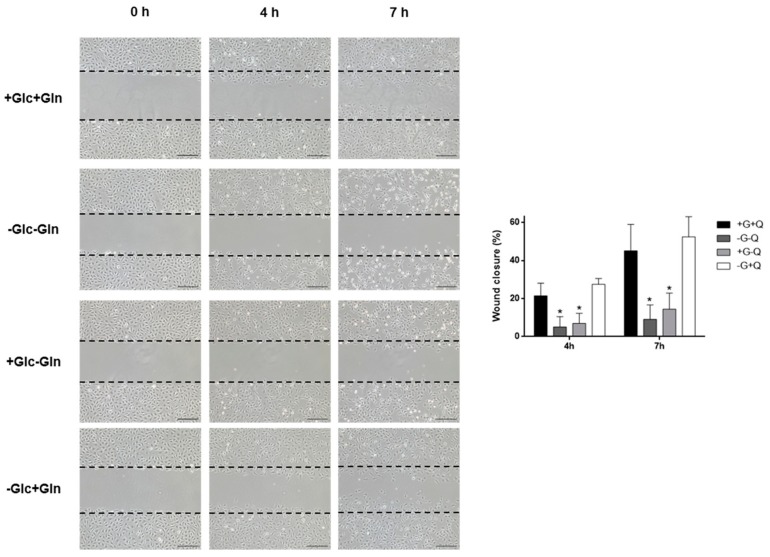
HMEC migration under different nutritional conditions. Representative images and quantification of wound closure of cells incubated in the presence or absence of 5 mM glucose (G) and/or 0.5 mM glutamine (Q) for 0, 4, and 7 h (scale bar = 200 µm). Data are expressed as means ± SD of three independent experiments. **p* < 0.05 versus condition with glucose and glutamine.

**Figure 4 biomolecules-09-00332-f004:**
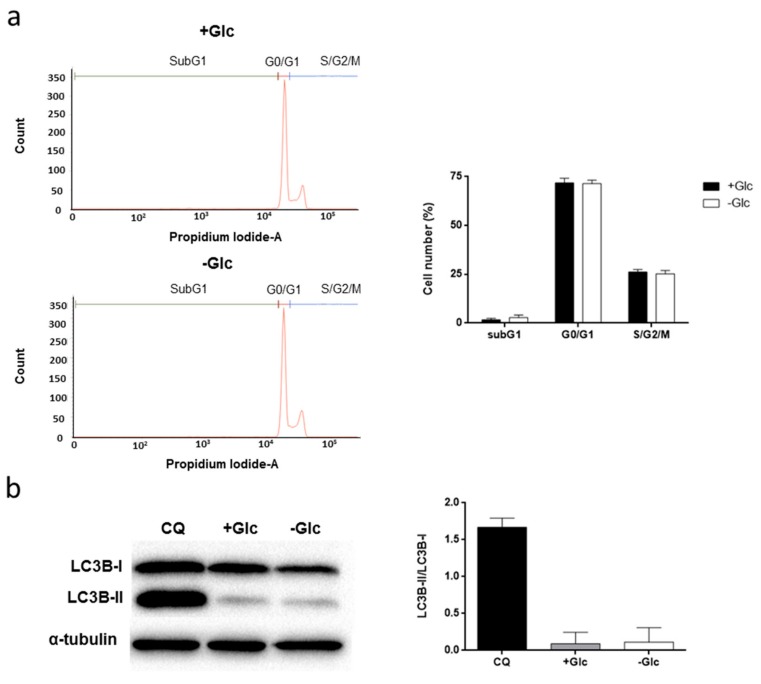
Effect of glucose starvation on cell cycle distribution and autophagy. (**a**) Cell cycle distribution of subpopulations of HMEC grown in the presence or absence of 5 mM glucose for 48 h, stained with propidium iodide and percentage of subG1, G1 and S/G2/M cells were determined using a FACS VERSE^TM^ cytometer. (**b**) LC3B-I and LC3B-II protein expression were measured by Western blot in cells grown in the presence or absence of 5 mM glucose. A positive control incubating cells with 5 mM glucose and 50 μM chloroquine (CQ) for 16 h was included. Data from Western blot are normalized against α-tubulin expression. Data are expressed as means ± SD of three independent experiments. Glutamine was added to the media in all experiments.

**Figure 5 biomolecules-09-00332-f005:**
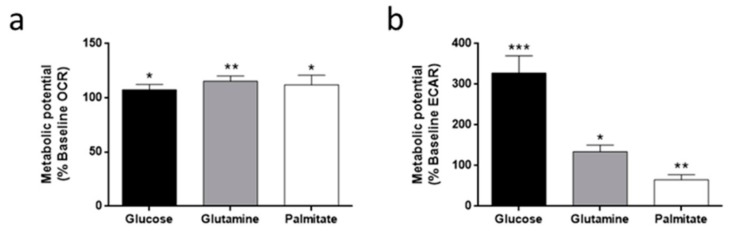
Flux analysis in HMEC. (**a**) Oxygen consumption rate (OCR) and (**b**) extracellular acidification rate (ECAR) in the presence of 5 mM glucose, 0.5 mM glutamine or 0.5 mM palmitate. Data are expressed as means ± SD of three independent experiments with triplicate samples each. **p* < 0.05, ***p* < 0.01, ****p* < 0.001 versus control without any metabolic substrate.

**Figure 6 biomolecules-09-00332-f006:**
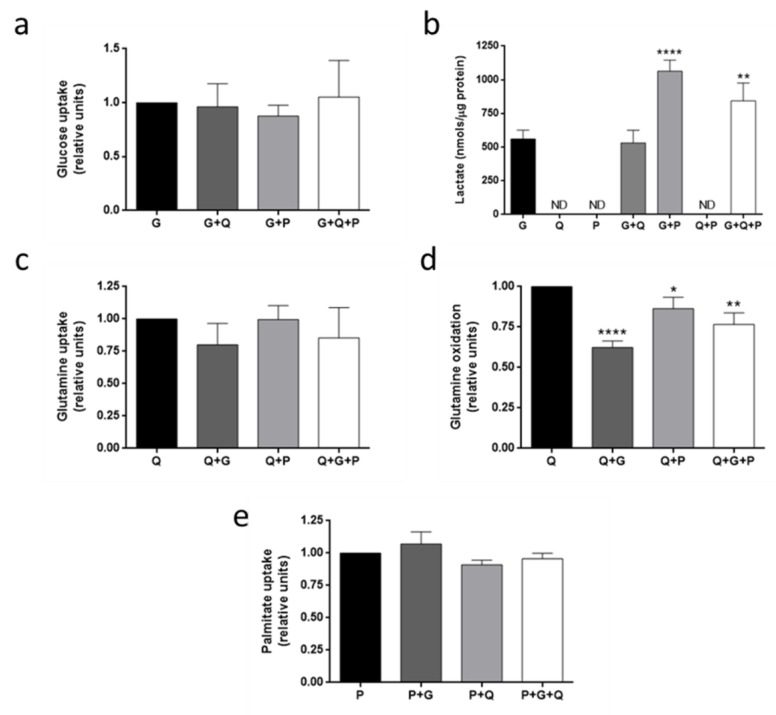
Substrate uptake and utilization in HMEC. (**a**) Glucose uptake, (**b**) lactate production, (**c**) glutamine uptake, (**d**) glutamine oxidation and (**e**) palmitate uptake were determined in HMEC in the presence or absence of 5 mM glucose (G), 0.5 mM glutamine (Q) and/or 0.5 mM palmitate (P). Data are expressed as means ± SD of three independent experiments. **p* < 0.05, ***p* < 0.01, *****p* < 0.0001 versus glucose (**a**,**b**), glutamine (**c**,**d**) or palmitate (**e**) conditions. ND: non detected.

**Figure 7 biomolecules-09-00332-f007:**
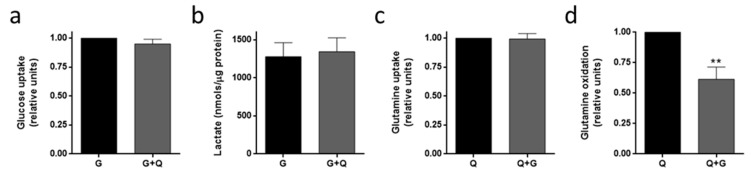
Substrate uptake and utilization in HUVEC. (**a**) Glucose uptake, (**b**) lactate production, (**c**) glutamine uptake and (**d**) glutamine oxidation were determined in HUVEC in the presence or absence of 5 mM glucose (G) and/or 0.5 mM glutamine (Q). Data are expressed as means ± SD of three independent experiments. ***p* < 0.01 versus glutamine condition.

**Figure 8 biomolecules-09-00332-f008:**
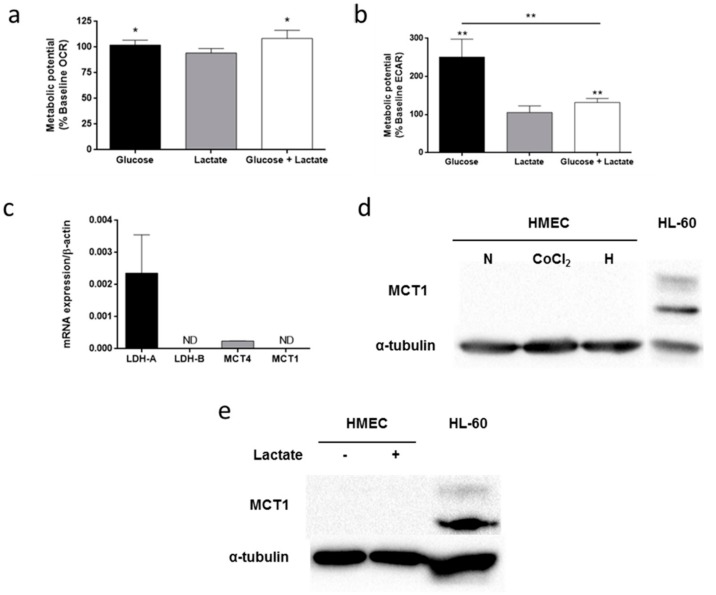
Lactate metabolism in HMEC. (**a**) Oxygen consumption rate (OCR) and (**b**) extracellular acidification rate (ECAR) were measured in the presence of 5 mM glucose and/or 10 mM lactate. (**c**) mRNA expression of LDH-A, LDH-B, MCT4 and MCT1. (**d**) MCT1 protein expression was determined under normoxia, 200 μM CoCl_2_ or 1% hypoxia. (**e**) MCT1 protein expression was determined in the presence or absence of 10 μM lactate. HL-60 under normoxia were used as positive control. Data are expressed as means ± SD of three independent experiments. **p* < 0.05, ***p* < 0.01 versus no glucose or lactate available (**a**,**b**). ND: non detected.

**Table 1 biomolecules-09-00332-t001:** Primers used for qPCR.

Gene	Primers	Annealing Temperature (°C)	Amplicon Size (bp)
β-actin	Fw: GACGACATGGAGAAAATCTG Rv: ATGATCTGGGTCATCTTCTC	60	131
LDH-A	Fw: CACCATGATTAAGGGTCTTTAC Rv: AGGTCTGAGATTCCATTCTG	60	87
LDH-B	Fw: CAACAATGGTAAAGGGGATG Rv: TCACTAGTCACAGGTCTTTTAG	60	189
MCT1	Fw: GAGGTCCTATCAGCAGTATC Rv: CAATGACTCCAATACAGACG	60	144
MCT4	Fw: CAGTTCGAGGTGCTCATGG Rv: ATGTAGACGTGGGTCGCAT	60	140

Note: LDH-A (lactate dehydrogenase A); LDH-B (lactate dehydrogenase B); MCT1 (monocarboxylate transporter 1); MCT4 (monocarboxylate transporter 4).

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
