# Peer review of "Highly Glycolytic Immortalized Human Dermal Microvascular Endothelial Cells are Able to Grow in Glucose-Starved Conditions"

_biomolecules, 2019, doi:10.3390/biom9080332_

Reviewer 1 Report

This manuscript by Ocana characterized the metabolic profile and fuel dependence of the human dermal microvascular endothelial cells in the presence or absence of glucose, glutamine, paimitate, or lactate. The authors demonstrated that HMEC was highly glycolytic. However, HMEC required glutamine instead of glucose for sustained cell proliferation. Glucose deprivation neither induced autophagy nor altered cell cycle. In addition, HMEC did not express MCT1 under either normoxic or hypoxic condition, thus limiting it ability of uptaking extracellular lactate. This study provides new insight to the understanding of metabolic profile of HMEC and EC-related diseases.

Broad comments:

1.     It is novel to study the metabolic profile and fuel dependence of HMEC. However, a comparison and discussion about the metabolism of ECs from other human microvascular beds could provide more value to the paper. Using BAEC as a macrovascular endothelial cell line is not adequate since it is from a different species.

2.     In the abstract, the author could not draw the second main conclusion based on the date since they did not compare to the ECs from other tissue origins. It could be implication.

3.     The authors mentioned that the HMEC used in this paper was impportalized. Please provide more information about how this cell line was generated and whether immortalization affect the metabolism of cells.

4.     Since HUVEC was not tested in this paper, information about HUVEC could be removed.

5.     A brief, but more detailed description of the “2.12 RNA isolation and purification and cDNA synthesis” could be more helpful.

6.     Statistical analysis for Figure 2, Figure 5, Figure 6b should use one-way ANOVA with post hoc test.

Specific comments:

1.     Line 60, spell out the full name for MCT1 when first mentioned in the text.

2.     Line 107, typo: “Cells were counted every day for 5 fives…”?

3.     Line 151, typo: Radiactive?

4.     In Figure 1d, there are only three groups (+Glc+Gln, -Glc-Gln, and -Glc+Gln). +Glc-Gln as indicated by the legend was not seen in the figure.

5.     Line 226, where is Figure 1e?

Author Response

Please see uploaded pdf file containing the complete reply to Reviewer 1.

Reviewer 2 Report

The present study uses endothelial cells of different origin to test their dependence on specific metabolic substrates for growth. While the concept of the study is interesting, we feel that some considerations should be made:

General comments

·    In the discussion section, the authors mention that the microvascular EC line they have used has been immortalised. It is questionable whether an immortalized microvascular cell line is suitable for metabolic studies as they are presented here. It is likely that the immortalization (not described) affects signaling pathways which highly affect and reprogram EC metabolism and ECs might therefore depends on other nutrient sources. Therefore, ideally it would need to be validated in primary microvascular ECs whether the same effects of glutamine and glucose starvation are observed in these cells. Only then, it would be interesting to draw microvascular-specific conclusions. If not feasible, the authors should consider discussing this possible drawback in the discussion section.

·     Cell growth and EdU incorporation were used as a measure of proliferation. However, the process of angiogenesis also involves e.g. the migratory capacity of ECs. This is not covered in this paper but should at least be mentioned/discussed. Ideally, sprouting angiogenesis assays for different withdrawal combinations of metabolic substrates are performed.

·     Glutamine starvation appears to be different between microvascular and macrovascular ECs. This difference should be shown in the main Figures and this should be discussed more extensively.

·     Discuss the findings of this paper in the light of previously obtained results (Huang et al. 2017 EMBO J).

·     Hypoxia experiments are a bit disconnected from the paper. Consider  moving to the supplemental section?

Minor comments

Abstract

·     Editing of English language style required

o  Phrase the following differently

§ Line 13: ECs are responsible of angiogenesis àECs form the inner lining of blood vessels

§ Line 14-15: An exacerbated, pathological angiogenesis occurs in several diseases, including cancer àExcessive angiogenesis is a hallmark of several diseases, including cancer

§ Line 15: The study has become an emerging issue à[the study itself is not an issue; indicate that the number of studies on EC metabolism has expanded/increased over the last years.

§ Line 17-18: long- term and short- term à[remove, because this raises questions of what is meant by long or short term; this needs to be explained in detail in the main text] 

§ Line 18: metabolite dependence à[why do you not use the term “substrate dependence”, just like it has been done for the main text?]

§ The role of only 3 main substrates were studied, please indicate in abstract which ones.

§ Line 21: “metabolite” à“substrate”

§ Line 25: unexpected situations à[this term is too vague, indicate in which conditions ECs show metabolic plasticity]

§ Line 24-26: the conclusions are all formulated using the word “may”, suggesting that it is not possible or still unsure to draw conclusions from the obtained results. Make a statement where you can (e.g. the results indicate that   

·     Conceptual issues

o  Please provide a rationale why dermal endothelial cells were used

o  Please provide a rationale why microvascular endothelial cells were used

§ Metabolic differences in microvascular versus macrovascular ECs are relevant. Elaborate on those differences and what is already known. More experiments, comparing other macrovascular cell types to MVECS, are recommended to show these metabolic differences.

o  Indicate clearly that an immortalized cell line was used

§ To prevent the reader from assuming that primary cells were used

o  Lactate uptake was never tested. Do not make the statement that cells are “unable to take up lactate” and certainly remove it from the title.

Introduction

·     Line 32: Use another citation

·     Line 32: “highly regulated in healthy individuals” à[do you suggest that it is not highly regulated in pathological angiogenesis; rephrase]

·     Line 33: “pathological activation” à[be more clear] excessive blood vessel growth/excessive angiogenesis

·     Line 35: “other pharmacological targets” à[this is vague, indicate what you mean, any examples?]

·     Line 35-36: remove: “Endothelial cells (ECs) are responsible of the angiogenic process.”

·     Line 38: “relevant” à“essential”

·     Line 40: remove: “in general”

·     Line 41: has been considered a potential target for àhas been considered asa potential target for

·     Line 41: angiogenic-related diseases àangiogenesis-related diseases

·     Line 42: Repetitive

·     Use the introduction to introduce the rationale for the use of dermal and microvascular ECs

·     It is indicated that it has been shown that ECs are highly glycolytic. De Bock et al. have shown that also microvascular ECs are highly glycolytic (Cell, 2013). Still it is mentioned that some works (plural) show the opposite, however, only one reference is cited. àLine 48

·     Line 47: rephrase “a high glycolytic rate is given(?) in HMECs”

·     Line 50-51: rephrase “look into the immediate appetite(?) for each…”

·     It is claimed that physiological concentrations of the metabolic fuels are used. Elaborate on why which concentrations were used, where this has been published and in which settings (healthy versus diseased (cancer?); mention that these settings are different: diabetes <-> glucose; cancer <-> lactate, etc…)

·     Line 60-61: as long as we are concerned àas far as we are aware

·     Line 57-60: break up in individual sentences. Now the sentence is too long, creating confusion as to which factor influences another one – or not.

·     Line 61: use other word for “just”

·     Have you verified that MCT1 expression has been determined specifically in microvascularECs in the CNS?

Methods

Why was 14C-labeled lactate never used to measure lactate uptake – please do so.

Results

·     Figure 1B-D: Make +Glc-Gln visible on graph, or display in another way

·     Line 202: rephrase “huge nutrient dependency”

·     Line 205: noticeably àimportantly

·     Line 226: where is Figure 1e?

·     Line 228: remove “that”

·     Line 228: inexistent à[it is not inexistent, give percentages in text]

·     Line 242: increased àincrease

·     Line 250-252: only ECAR was used for the statement that these cells are highly glycolytic. There are better techniques available to measure the flux from extracellular glucose through the glycolytic pathway. Therefore, it is recommended to use 3H-5-D-Glucose. Also to make the statement stronger that these cells are highly glycolytic, it would be better to compare them directly with other cells known to be highly glycolytic to validate this point.

·     Line 263 (idem 301): Figure potencial àpotential

·     Line 274: text says undetectable but the numbers are negative – please indicate N.D. in graph.

·     Line 287: Indicate lactate concentration.

·     Line 292: a positive control should be provided for the detection of LDH-B and MCT1 mRNA. 

·     Protein levels for MCT1 are now shown in HL-60 cells as a positive control, but there is no explanation for why these cells were chosen. Since it is stated in the text that MCT1 is expressed at the protein level in HUVECs, why was this not shown in these cells?

·     Line 293: phrase is not correct English.

·     Line 301: Figure 6d: the alpha-tubulin blot is not cut, whereas this is done for MCT1. How can one be sure that these blots correspond? Figure 6e, loading control can be improved.

Discussion

·     Line 322: Can you speculate on why physiological palmitate levels in cells would induce toxicity for prolonged periods of time? In vivo this would not be the case.

·     Line 327: “for several days”; if this statement is made experiments with more timepoints should be performed to show when cells stop growing and it should be indicated exactly how many days cells can function without glucose but with glutamine.

·     Line 341: It seems like an overstatement that HMECs are dependent on glutamine, but not glucose to grow. During the first days, it seems that they are not dependent on glucose for growth, but after that they are dependent on glucose. Therefore, it could well be that during the first days compensation mechanisms come into play that keep the cells growing. Was the possibility of glycogen storage and subsequent glycogenolysis during glucose withdrawal considered? Elaborate on this.

·     Line 378-388: The fact that immortalized cells have less strict nutritional requirements already indicates that it might not be a valid model for human microvascular ECs at a metabolic level.

·     Line 397-398: As mentioned, more experiments should be included comparing HMECs to macrovascular cells, because this is an interesting point

·     Line 308: versus glucose? Why is there an asterisk on the glucose bar?

·     Line 315: It is claimed that in most of these works non-physiological concentrations were used. Please indicate which concentration in which publications and check whether this was maybe useful for the setting in which they chose to study EC metabolism (e.g. high lactate in a tumor setting).

Author Response

Please, see uploaded pdf file containing the complete reply to Reviewer 2.

Round  2

Reviewer 2 Report

The manuscript has been improved by making textual changes proposed by us.

Unfortunately, some radioactive metabolic experiments could not be performed, given the limited time-frame for the revision. However, the newly performed experiments and changes made are sufficient. We feel that - if the following adaptations can be made to the manuscript - publication in the Biomolecules journal is justified; there would be no need for us to review the revised manuscript again.

34 - [Adapt to:] However, excessive pathological angiogenesis has been described in some diseases such as …

39 - [Adapt to:] The study of endothelial cell (EC) metabolism grew in importance…

223 - [Adapt to:] Instalation —> Installation

376 - [The order of the pictures should match the order of the graphs, please adapt]

568 - [Fig. 8d - a separation between HMEC and HL-60 is present for MCT1, please do this also for alpha-tubulin]

692 - [Adapt to:] Other less well characterized lactate transporters, such as MCT2, have been…

774 to 777 - [add references]

Author Response

RESPONSE TO REVIEWER 2’S COMMENTS/SUGGESTIONS

General comments

The manuscript has been improved by making textual changes proposed by us. Unfortunately, some radioactive metabolic experiments could not be performed, given the limited time-frame for the revision. However, the newly performed experiments and changes made are sufficient. We feel that - if the following adaptations can be made to the manuscript - publication in the

Biomolecules journal is justified; there would be no need for us to review the revised manuscript again.

34 - [Adapt to:] However, excessive pathological angiogenesis has been described in some diseases such as …

39 - [Adapt to:] The study of endothelial cell (EC) metabolism grew in importance…

223 - [Adapt to:] Instalation —> Installation

376 - [The order of the pictures should match the order of the graphs, please adapt]

378 - [Fig. 8d - a separation between HMEC and HL-60 is present for MCT1, please do this also for alpha-tubulin]

479 - [Adapt to:] Other less well characterized lactate transporters, such as MCT2, have been…

774 to 777 - [add references]

Thank you very much for your overall positive evaluation and your valuable comments and suggestions. We have prepared a second amended version of our manuscript taking all your suggestions into account. We have found some other minor mistakes and misspelling, which have also amended in the new version of the manuscript.

Thanks again to Reviewer 2 for his/her carefully revision of this manuscript. We think that they have been very helpful to improve our work.